# Deconstructing the Inductive Biases of Hamiltonian Neural Networks

**Nate Gruver, Marc Finzi, Samuel Stanton, Andrew Gordon Wilson**
New York University

## Abstract

Physics-inspired neural networks (NNs), such as Hamiltonian or Lagrangian NNs, dramatically outperform other learned dynamics models by leveraging strong inductive biases. These models, however, are challenging to apply to many real world systems, such as those that don't conserve energy or contain contacts, a common setting for robotics and reinforcement learning. In this paper, we examine the inductive biases that make physics-inspired models successful in practice. We show that, contrary to conventional wisdom, the improved generalization of HNNs is the result of modeling acceleration directly and avoiding artificial complexity from the coordinate system, rather than symplectic structure or energy conservation. We show that by relaxing the inductive biases of these models, we can match or exceed performance on energy-conserving systems while dramatically improving performance on practical, non-conservative systems. We extend this approach to constructing transition models for common Mujoco environments, showing that our model can appropriately balance inductive biases with the flexibility required for model-based control.

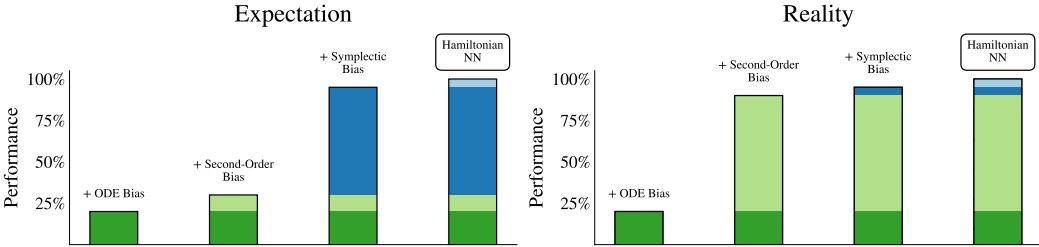

Figure 1: The common perception in physics-informed machine learning is that increased performance is the result of complex biases. We find, however, that simpler implicit biases (such as second-order structure) often account for almost all of the improvement over baselines.

## 1 Introduction

The inductive biases of convolutional neural networks, such as translation equivariance, locality, and parameter sharing, play a key role in their generalization properties. Other biases have similarly guided the development of deep models for other domains like graphs and point clouds. Yet since models often encode multiple biases simultaneously, it can be challenging to identify how each contributes to generalization in isolation. By understanding which inductive biases are essential, and which are merely ancillary, we can simplify our models, and improve performance. For example, Wu et al. (2019) demonstrate that Graph Convolutional Networks can be radically simplified to mere logistic regression by removing nonlinearities, leading to dramatic gains in computational efficiency, while retaining comparable accuracy.

Hamiltonian Neural Networks (HNNs) (Greydanus et al., 2019) have emerged as a leading approach for modeling dynamical systems. These dynamics models encode physical priors and outperform alternative Neural ODE approaches (Chen et al., 2018). In the spirit of Wu et al. (2019), we seek to identify the critical components of HNNs. Since Lagrangian models (LNNs) (Lutter et al., 2019a;

Cranmer et al., 2020) share the same structure and inductive biases as HNNs, we focus on HNNs where energy conservation and symplecticity are more explicit.

HNNs encode a number of inductive biases that help model physical systems:

1. **ODE bias**: HNNs model derivatives of the state rather than the states directly.
2. **Second-order (SO) bias**: HNNs model changes in position through changes in velocity.
3. **Energy conservation bias**: HNNs conserve their learned energy function.
4. **Symplectic bias**: HNNs dynamics are *symplectic*: phase space areas are conserved, and the vector field has a Hamiltonian structure.

In this paper we theoretically and empirically examine the role of these biases. In contrast to conventional wisdom, we find that the generalization benefit of HNNs is not explained by their symplectic or energy-conserving properties, but rather by their implicit second-order bias. We highlight these findings in Figure 1. Abstracting and extracting out this second-order bias, we show how to improve the performance of Neural ODEs, empowering applications when HNNs assumptions are violated as is often the case. Code for our experiments can be found at: https://github.com/ngruver/decon-hnn.

## 2 RELATED WORK

**Physics-inspired models and energy conservation**   Since Greydanus et al. (2019), many researchers have sought to extend the HNN approach to make it more general or applicable to systems that break energy conservation. Jin et al. (2020), Li et al. (2020), Tong et al. (2021), and Xiong et al. (2020) propose methods for creating neural networks that preserve symplectic structure directly. Zhong et al. (2020), and later Desai et al. (2021), Li et al. (2021), and Lee et al. (2021) propose models with additional capacity for changes to the system energy. In our analysis, we compare against approaches that add additional capacity for changes in energy, although our approach is fundamentally different — we attempt to remove unnecessary biases rather than add complexity.

**Physics-inspired models for control**   A large and rapidly expanding body of work explores how to use physics-based biases in dynamics models with controls or contacts (Lutter et al., 2019b; Zhong et al., 2019; Gupta et al., 2019; 2020; Chen et al., 2019; Hochlehnert et al., 2021; Chen et al., 2020; Zhong et al., 2021b) intended for application in model-based planning. Especially relevant to our evaluations, Alvarez et al. (2020) models MuJoCo (Todorov et al., 2012) trajectories with a numerically integrated neural network but does not explore other physics-inspired inductive biases.

**Analyzing physics-based inductive biases**   Karniadakis et al. (2021) and Liu et al. (2021) propose conceptual frameworks for deep learning with physics-based inductive biases, but present minimal empirical analysis of common design decisions. Zhong et al. (2021a) compare many approaches to physics-inspired deep learning, with results that parallel some findings here. The contribution of our work, however, is not simply benchmarking but also an actionable theory of HNNs' success. In spirit, our work is similar to Botev et al. (2021), which also critically examines the role of physics-based priors in dynamics models, though their focus is on learning latent space dynamics, while we focus on temporal models in observation space.

## 3 BACKGROUND

We consider dynamical systems described by ordinary differential equations (ODEs) which determine how the system evolves over time. Even with high order derivatives, these systems can be arranged into the form $\frac{dz}{dt} = F(z, t)$ for $z \in \mathbb{R}^n$. If the dynamics $F$ are time-independent, then the ODE can be understood as a *vector field*, specifying at each point where the next state will be.

Neural ODEs (NODEs) (Chen et al., 2018) parametrize a vector field with a neural network and learn the dynamics directly from observed trajectories. A NODE dynamics model $\hat{F}_\theta$ is rolled out from the initial condition $z_0$ with ODE integration, $\hat{z}_t = \text{ODESolve}(z_0, \hat{F}_\theta, t)$, and fit to trajectory data by minimizing an L2 loss $L(\theta) = \sum_{t=1}^{T} \|\hat{z}_t - z_t\|^2$ between the predicted and observed trajectories, $\hat{z}_t$ and $z_t$.

Like NODEs, HNNs also model dynamical systems as a parameterized vector field,

$$\frac{dz}{dt} = J\nabla \hat{H}_\theta \text{ with } J = \begin{bmatrix} 0 & I \\ -I & 0 \end{bmatrix}, \tag{1}$$

where $\hat{H}_\theta$ is a neural network with scalar output (Greydanus et al., 2019).

This differential equation expresses Hamiltonian dynamics where $p$ and $q$ are the canonical positions and momenta,

$$z = \begin{bmatrix} q \\ p \end{bmatrix} \text{ and } \frac{d}{dt}\begin{bmatrix} q \\ p \end{bmatrix} = \begin{bmatrix} \frac{\partial \hat{H}}{\partial p} \\ -\frac{\partial \hat{H}}{\partial q} \end{bmatrix}. \tag{2}$$

It is common practice to also explicitly define $H = T + V$, where $T$ and $V$ are the kinetic and potential energy (Zhong et al., 2019; Gupta et al., 2019; Finzi et al., 2020). The assumption that the Hamiltonian is *separable* holds for mechanical systems and allows for further simplification,

$$\hat{H}_\theta(q,p) = \frac{1}{2}p^T M_\theta^{-1}(q)p + V_\theta(q), \tag{3}$$

where positive definite mass matrix $M$ and scalar $V$ are outputs of the neural network.

## 4 BREAKING DOWN HNN PERFORMANCE

In the following section we investigate to what extent commonly held beliefs about HNN properties actually explain the ability of HNNs to generalize. To separate out the different properties of these models, we select synthetic environments from Finzi et al. (2020) and Finzi et al. (2021) that are derived from a time independent Hamiltonian, where energy is preserved exactly. We use kChainPendulum, a $k$ link pendulum in angular coordinates, and kSpringPendulum, a pendulum connected with $k$ spring links in Cartesian coordinates.

We compute relative error between predicted states, $\hat{z}$, and ground truth, $z$, as $\|\hat{z} - z\|_2/\|\hat{z}\|_2\|z\|_2$ and between the predicted and ground truth Hamiltonian as $|\hat{H} - H|/|\hat{H}||H|$. Following Finzi et al. (2020), we evaluate the performance of the model by computing the geometric mean of the relative error of the state over rollouts of length 20 times the size used at training, which more faithfully predicts downstream performance compared to other metrics like MSE (Finzi et al., 2020). The geometric mean of a function $f$ over time $T$ is given by $\exp(\frac{1}{T}\int_{t=0}^{T}\log f(t)dt)$.

### 4.1 ENERGY CONSERVATION

HNNs are commonly believed to be superior for energy conservation than comparable models (Greydanus et al., 2019; Cranmer et al., 2020). While there is some empirical evidence to support this claim, a precise mathematical explanation for why this should be the case has not been established. Surprisingly, we find that conditioned on the trajectory reconstruction error, HNNs are no better at conserving the true energy of the system than unconstrained NeuralODE models.

Up to the numerical accuracy of an ODE solver, HNNs do conserve their own learned energy function $\hat{H}$ (different from the true energy of the system $H$), due to basic properties of Hamiltonian mechanics,

$$\frac{d\hat{H}(\hat{z})}{dt} = \nabla\hat{H}^\top\frac{d\hat{z}}{dt} = \nabla\hat{H}^\top J\nabla\hat{H} = 0, \tag{4}$$

where the last equality follows from the antisymmetry of $J$. If the HNN has fit the data well, we might expect $\hat{H}$ to be close to the true Hamiltonian $H$, and so if $\hat{H}$ is conserved then it seems that $H$ should be too. As we show in Appendix B.2, the problem with this argument is that we have no guarantees that $\hat{H}$ and $H$ are close even if we have fit the data well with low error in the rollouts or the dynamics. Since the dynamics error and the training rollout error depend only on gradients $\nabla\hat{H}$, while the gradients may be close, the differences between the two scalar functions can grow arbitrarily large.

In Figure 2 (left) we show that the degree of energy conservation is highly correlated with the rollout error of the model, regardless of the choice of architecture. While HNNs have better rollout

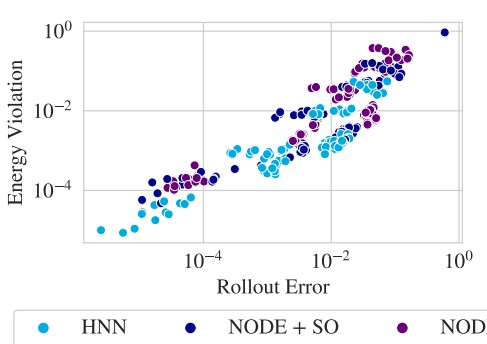 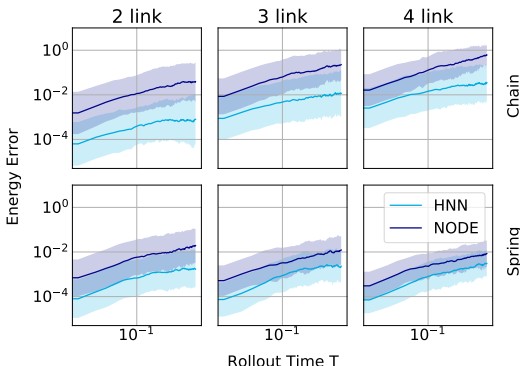

Figure 2: **Left**: The degree of energy violation ($|\hat{H} - H|/|\hat{H}||H|$) on test rollouts as a function of rollout relative error ($\|\hat{z} - z\|/\|\hat{z}\|\|z\|$) across different environments and random seeds. Both HNNs and NeuralODEs are scattered around the line $x = y$. Conditioned on the rollout performance, whether or not the model is Hamiltonian has little impact on the energy violation. **Right**: Energy violation on test trajectories is plotted as a function of the time $T$ of the rollout, with the shaded regions showing 1 standard deviation in log space taken across 5 random seeds and the test trajectories.

performance than NeuralODEs, they are on the same regression line with Rollout Error $\propto$ Energy Violation. The differences in energy violation are best predicted by the rollout error and *not* the architecture.

In Appendix B.1, we derive a bound that helps explain this behaviour. Given that the trajectories are in a bounded region of the phase space and that there is a fixed amount of error in the dynamics model, the energy violation grows at most linearly in time and the dynamics error. In Figure 2 (right) we demonstrate empirically that energy is *not* conserved as time progresses for HNNs.[1] In fact, the energy error of both NODE and HNN models grow linearly as our bound suggests. Although energy conservation may be helpful for generalization, the evidence does not indicate that HNNs are inherently better at conserving energy than NODEs, suggesting that the superior generalization of HNNs cannot be attributed to superior energy conservation.

## 4.2 SYMPLECTIC VECTOR FIELDS

A defining property of Hamiltonian mechanics is the fact that the dynamics are *symplectic*. If energy conservation does not explain the effectiveness of HNNs, the symplectic property of HNN dynamics may be the cause. Informally, one of the consequences of symplectic dynamics is that every part of the state space is equally attractive and repulsive. There are no sources where all nearby trajectories flow out from, or sinks where all nearby trajectories flow into, only saddle points and centers where the inflow and outflow is balanced. Symplectic integrators (Leimkuhler and Skeel, 1994) make use of this property for more stable integration of Hamiltonian systems over very long timespans, and it is intuitive that enforcing this property in learned dynamics would have benefits also, at the very least for reducing the size of the hypothesis space.

More formally, symplecticity is the property that the $J$ matrix (the symplectic form) is preserved by the dynamics (Equation 1). This condition can be expressed as a constraint on the Jacobian $DF$ of the vector field $F(z) = \frac{dz}{dt}$ (here the derivative $D$ maps from a function to its Jacobian). In terms of the Jacobian, symplecticity is the condition $DF^\top J + JDF = 0$ or equivalently $(JDF)^T = JDF$ since $J^\top = -J$. The significance of this condition is that areas occupied by states in phase space (which have units of energy) are preserved by symplectic transformations. One consequence is that a volume of solutions in phase space will continue to occupy the same volume over time and will not be compressed or expanded. In other words, the vector field has 0 divergence: $\mathrm{Tr}(DF) = 0$, which one can derive from the above expression.

---

[1]The energy violation is considerably larger than merely the numerical error associated with the solver.

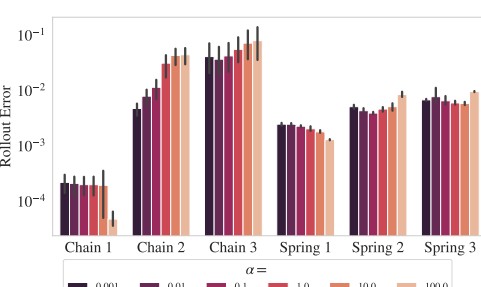 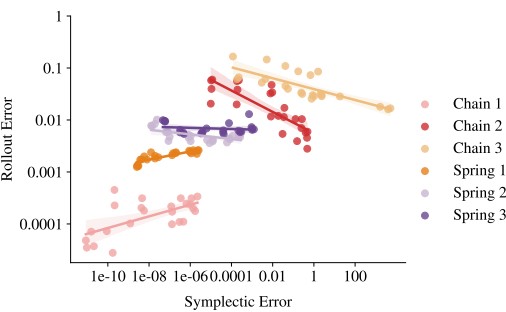

Figure 3: **Left**: Test rollout error as a function of the regularization weighting in the loss. Even at an optimally chosen symplectic regularization strength, the benefit to model generalization is negligible. **Right**: Test rollout error plotted against the final value of the symplecticity error for the regularized models. For systems with more than a couple degrees of freedom, symplecticity error is *negatively* correlated with the quality of predictions.

On $\mathbb{R}^n$, it can be shown that all symplectic vector fields can be expressed as the dynamics of some Hamiltonian system, and vice versa:

$$F = J\nabla H \iff (JDF)^T = JDF \qquad (5)$$

To show the forward direction, one can simply substitute in Hamiltonian dynamics. It is clear that the symplecticity property is satisfied: using $J^2 = -I$, $JDF = JD(J\nabla H) = J^2\nabla^2 H = -\nabla^2 H$, and $-\nabla^2 H$ is a symmetric matrix. The reverse direction is less obvious; however, by Poincaré's lemma if a vector field $F$ satisfies $(JDF)^T = JDF$ on $\mathbb{R}^n$, then there exists a Hamiltonian function $H$ such that $F = J\nabla H$, which is shown in subsection B.3.

The equivalence of Hamiltonian dynamics and symplecticity allows us separate the unique properties of HNNs from other inductive biases that result indirectly from modeling $F$ through $H$. Following Ghosh et al. (2020), we can create a regularizer $\text{SymplecticError}(F) = \|(JDF)^T - JDF\|^2$ that directly measures the degree to which the symplectic property is violated. By parametrizing a NeuralODE and regularizing the symplectic error, we can enforce Hamiltonian structure while still directly modeling $\frac{dz}{dt} = F$ rather than $H$. Alongside an unregularized NeuralODE, we can isolate and evaluate the benefit of this Hamiltonian structure bias with a direct comparison.

Surprisingly, we find that the Hamiltonian structure bias, as enforced by the symplectic regularizer, provides no real benefit to the model's ability to generalize over the long test rollouts (Figure 3 left). The achieved symplectic error Figure 3 (right) is not positively correlated with the final test rollout error of the model, and in some cases is even *negatively* correlated. Even when the symplectic error is very low and the symplecticity condition is enforced, there is no consistent improvement on the rollout generalization.

### 4.3 SECOND-ORDER STRUCTURE

If the superior performance of HNNs over NeuralODEs does not come from their better energy conservation properties, nor from the symplectic structure of the predicted vector field, what is the true cause?

In previous work, authors have used slightly different implementations of HNNs. One subtle improvement over the original work (Greydanus et al., 2019) comes from explicitly splitting the Hamiltonian as $\hat{H} = T + V = p^\top M(q)^{-1}p/2 + V(q)$ and modeling the mass matrix $M(q)$ and the potential $V(q)$ with separate neural networks rather than using a single neural network for $\hat{H}$ (Zhong et al., 2019). This splitting enforces a strong assumption about the functional form of the Hamiltonian that applies to mechanical systems that makes it easier to learn and extrapolate. Through Hamiltons equations $\hat{F} = J\nabla\hat{H}$, this splitting is in fact specifying the relationship between position and momentum $\frac{dq}{dt} = \frac{\partial H}{\partial p} = M^{-1}(q)p$, and that forces can only affect $\frac{dp}{dt}$.

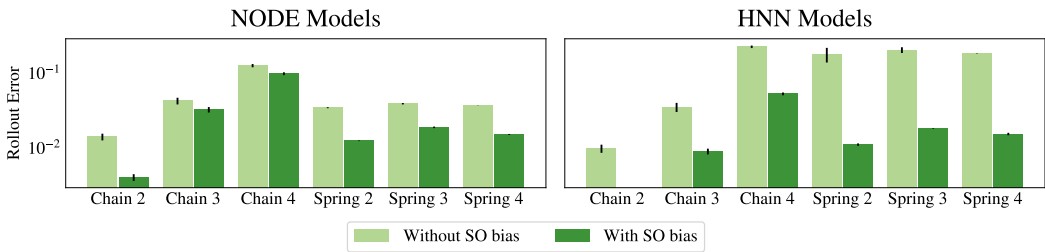

Figure 4: **Left**: NODE model with and without second-order structure (encoding $dq/dt = v$). **Right**: HNN models with and without second-order structure. Models with the SO bias significantly outperform those that do not. Error bars show standard error across 5 seeds.

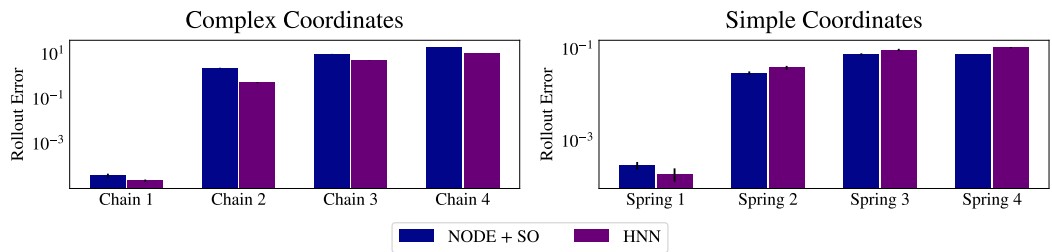

Figure 5: **Left**: Log rollout error for NODEs with second order bias and HNNs trained chain pendulums, where the analytic form of the Hamiltonian is simpler than the vector field. **Right**: Mechanics-sNNs and HNNs trained on spring pendulums, which have Hamiltonians and vector fields of similar complexity. HNNs outperform NODE with second order bias on systems that use non-Cartesian coordinates. Error bars show standard error across 5 seeds.

In fact, we can see that this assumption essentially leads to the second-order differential equation

$$\begin{bmatrix} \frac{dq}{dt} \\ \frac{dp}{dt} \end{bmatrix} = \begin{bmatrix} M^{-1}(q)p \\ -\frac{dV}{dq} \end{bmatrix} \implies \frac{d^2q}{dt^2} = \left( \frac{d}{dt} M^{-1}(q) \right) M(q) \frac{dq}{dt} - M^{-1}(q) \frac{dV}{dq} = A(q, \frac{dq}{dt}) \quad (6)$$

This second-order (SO) structure, $\begin{bmatrix} \frac{dq}{dt} \\ \frac{dv}{dt} \end{bmatrix} = \begin{bmatrix} v \\ A(q, v) \end{bmatrix}$, is a direct by-product of the separable Hamiltonian inductive bias, but is more general, applying to both conservative and non-conservative physical systems.

We can isolate the effect of this bias by directly observing its effect on both HNNs and NODEs. For HNNs the bias is made explicit in separable Hamiltonians, but not in the general case, when $H(q, p)$ is the direct output of the network instead of $V(q)$. We can design a NODE with second-order structure (NODE + SO) by setting $z = [q, p]$ with $p = Mv$ and $\frac{dq}{dt} = v$, $\frac{dp}{dt} = \tilde{A}_\theta(q, p)$, or equivalently just the second order equation $\frac{d^2q}{dt^2} = A_\theta(q, \frac{dq}{dt})$. Figure 4 shows the effect of second order structure on the test predictions of NODEs and HNNs. It is clear that this bias explains the superior performance of HNNs much more than other biases that are frequently given more credit. In fact, when we add this bias to NODE models, we see that their performance more closely matches HNNs than vanilla NODEs without creating a conservative or symplectic vector.

## 4.4 FUNCTIONAL COMPLEXITY

Adding second order structure to NODEs is always helpful, and matches the HNN performance for many of the systems. However, we see that there is still a gap for some systems, and curiously in each of these cases the system is described in a non-Cartesian coordinate system, such as with joint angles and Euler angles. Recall that with the symplecticity bias, we only found no benefit for enforcing that there *exists* a function $\hat{H}$ such that $\hat{F} = J\nabla\hat{H}$ bias while parametrizing $\hat{F}$; however, for HNNs this function not only exists, it is directly parametrized by the neural net. If the function

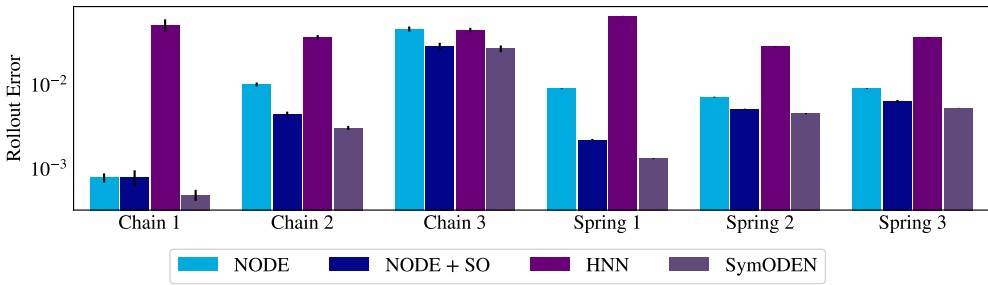

Figure 6: Comparing the performance on damped systems. The NODE + SO matches the performance of a SymODEN with a fraction of the parameters and compute. HNNs without forcing terms encode the wrong inductive biases and thus fit the data poorly. Error bars denote standard error across 5 seeds.

$\hat{H}$ happens to be a simpler function to express and learn than $\nabla H$, then representing the solution in this way can be beneficial.

For systems expressed in Cartesian coordinates like the spring pendulum, the mass matrix $M(q) = M$ is a constant, and so the gradients of $H = p^\top M p + V(q)$ are simple to express and learn. However, for systems with constraints such as the Chain pendulum, where states are typically expressed in angular coordinates, the mass matrix $M(q)$ will have a complicated form and that complexity will be magnified when taking the derivative. As an example, consider the Hamiltonian that an HNN must learn for the 2 link chain pendulum versus the vector field that a NODE + SO model must learn for this system (derived in Finzi et al. (2020), Appendix F.2). For the spring pendulum the functional complexity of the Hamiltonian and vector fields is comparable, while for the chain pendulum, the vector field contains many more terms.

Parameterizing such a system via its Hamiltonian simplifies the learning problem, and enables a neural network to converge more rapidly towards a plausible solution. This observation aligns with the insight in Finzi et al. (2020), which shows that changing the coordinate system to Cartesian dramatically simplifies the learning problem, at the expense of needing to enforce the constraints to the configuration space more directly.

Figure 5 shows the relative performance of the NODE+SO across the chain pendulum and spring pendulum environments. As we would now expect, the gap between the NODE+SO and HNN vanishes (and even favors NODE+SO) when complexity of the Hamiltonian and vector field are comparable.

### 4.4.1 NON-CONSERVATIVE SYSTEMS

Perfectly energy-conserving systems are useful for analyzing the limiting behaviour of physics-informed networks, but in the vast majority of real world applications, we do not model closed systems. Energy is changed through contact with the environment (as in friction or drag) or an actor applying controls. In these cases, HNNs can be generalized by adding a forcing term

$$\begin{bmatrix} \frac{dq}{dt} \\ \frac{dp}{dt} \end{bmatrix} = \begin{bmatrix} \frac{\partial H}{\partial p} \\ -\frac{\partial H}{\partial q} \end{bmatrix} + \begin{bmatrix} 0 \\ g(q,p) \end{bmatrix} u \tag{7}$$

as in SymODEN (Zhong et al., 2019; 2020) and Desai et al. (2021), where $u$ is the control input, which can be fixed as constant in systems with drag or friction.

Though SymODEN can accommodate controls and damping, we show that simply using the bias of second-order dynamics is sufficient to achieve nearly the same performance with much less complexity. We demonstrate the matching performance on our n-body pendulum systems, augmented with drag, $\frac{dp}{dt} = -\frac{\partial H}{\partial q} - \lambda v$ (Figure 6).

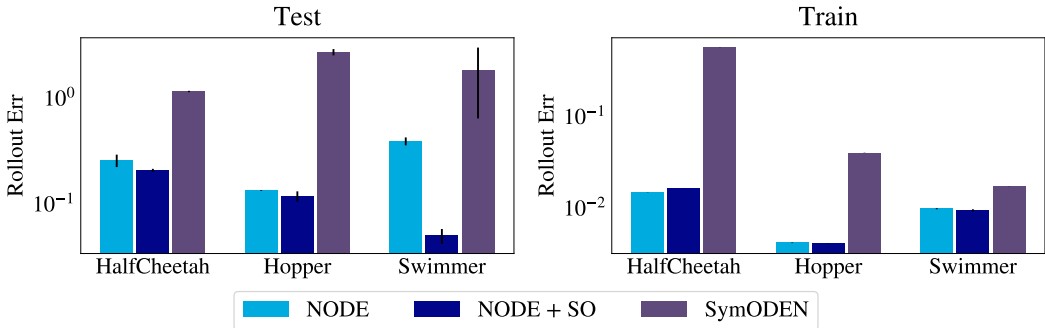

Figure 7: HNNs perform very poorly on complex dynamics like OpenAI Gym Mujoco control systems. Biasing the model towards Hamiltonian dynamics makes it difficult to fit the training data. Simply imposing second-order structure on a NODE is much more effective. Error bars show standard error across 4 seeds.

## 5 MODELING MUJOCO TRANSITION DYNAMICS

HNNs are typically evaluated on relatively simple systems, like those considered in the previous sections. In principle, we would expect these results to extend to more complex systems that are governed by similar physical laws. In practice, however, there is little evidence to suggest that applying HNN methods to complex systems is easy or effective.

The MuJoCo physics simulator (Todorov et al., 2012) is one such complex system that we would expect to benefit from HNN inductive biases. Gym Mujoco systems are heavily used in model-based reinforcement learning (MBRL) literature, but the dynamics models commonly used are surprisingly simple (often just MLPs trained to predict the next change in state) (Chua et al., 2018; Wang and Ba, 2019).

Given how much benefit other applications of deep learning have derived from specialized architectures, such as CNNs for computer vision (LeCun et al., 1989), RNNs for NLP (Mikolov et al., 2010), or WaveNets for audio (Oord et al., 2016), we would expect analogous improvements to be possible in MBRL. Algorithms for MBRL are infamously sensitive to choice of prediction horizon, and one possible explanation is poor generalization caused by weak inductive biases (Janner et al., 2019; Pan et al., 2020; Amos et al., 2021). Improving model design for mechanical systems has the potential to improve both the sample efficiency of MBRL algorithms and their robustness.

We train NODEs and HNNs on trajectories from several OpenAI Gym Mujoco environments (Brockman et al., 2016). Crucially, we compare NODEs endowed with second-order structure (NODE + SO) against pre-existing NODE and HNN models, as we did in Section 4.4.1. Note that with fixed step size integrators, NODEs are equivalent to discrete transition models that predict the next state or delta directly with an MLP, and therefore our NODE baseline is representative of models commonly used in MBRL. See Appendix C for implementation details.

In Figure 7 we show that NODE + SO significantly outperforms baseline methods. Surprisingly HNNs underperform NODEs on all the systems we consider. Although the HNN could in principle learn the dynamics, in practice the bias towards Hamiltonian dynamics makes fitting the training data very difficult and provides no tangible benefit to generalization. This outcome is notably different from what we observe in toy tasks, where HNNs can fit non-conservative systems (e.g., pendulums with drag) with little difficulty.

In spirit, our results parallel the findings of Wu et al. (2019) in the different setting of graph CNNs. We are able to distill the inductive biases of HNNs into a NODE + SO without losing performance on systems that HNNs perform well on, and, even more importantly, these reduced systems are much more capable of scaling to complex systems and larger datasets.

## 6   CONCLUSION

In this paper, we deconstructed the inductive biases of highly performing HNN models into their component parts, a NeuralODE, symplecticity, conservation of the learned energy function, and second order structure. Contrary to conventional wisdom, the success of HNNs is not from their energy conservation or symplecticity, but rather from the assumption that the system can be expressed as a single second order differential equation. Stripping away the other components of an HNN, we are left with a model that is simpler, more computationally efficient, and less restrictive in that it can be directly applied to non-Hamiltonian systems. As a consequence, we are able to apply the resulting model to constructing transition models for the challenging Mujoco locomotion environments, with promising performance.

### ACKNOWLEDGEMENTS

The authors would like to thank Greg Benton and Pavel Izmailov for helpful comments on our original submission. This research is supported by an Amazon Research Award, Facebook Research, Google Research, NSF I-DISRE 193471, NIH R01DA048764-01A1, NSF IIS-1910266, and NSF 1922658 NRT-HDR: FUTURE Foundations, Translation, and Responsibility for Data Science.

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

# A  APPENDIX OUTLINE

This appendix is organized as follows. In Section B, we present proofs for the energy conservation properties of HNNs and Neural ODEs, as well as a more detailed description of symplecticity. In Section C, we described the details of our experiments on Mujuco, including our data preprocessing, training, and architectural decisions. Lastly, in Section D, we provide additional experimental results requested by reviewers of our original submission. These include a comparison of alternative loss functions and a comparison on additional rigid body systems.

# B  MATHEMATICAL DETAILS

## B.1  ENERGY CONSERVATION FOR NEURAL ODES

Let $F = J\nabla H$ be the ground truth dynamics of a time independent Hamiltonian, and $\hat{F}$ be the dynamics learned by a neural network through an learned Hamiltonian $\hat{H}$ for HNNs or otherwise. Given some initial condition $z_0$, let $\hat{z}_T$ denote the solution to $\dot{z} = \hat{F}(z)$ at time $T$ starting from $z_0$ and $z_T$ be the solution to the ground truth dynamics from $z_0$.

Suppose the error in the dynamics model $e(z) = \hat{F}(z) - F(z)$ is bounded $\forall z : \|e(z)\| < \delta$ and that we are only considering a bounded region of the state space (such as the states of a pendulum with bounded energy).

Since energy is conserved $H(z_t) = H(z_0) = H(\hat{z}_0)$, we can write the the energy error

$$H(\hat{z}_T) - H(z_T) = H(\hat{z}_T) - H(\hat{z}_0) = \int_{\hat{z}_0}^{\hat{z}_T} \nabla H(z)^\top dz$$

Since the value is independent of the path, we may consider the path given by the approximated dynamics $\hat{z}_t$. Noting that $d\hat{z}/dt = \hat{F}(\hat{z}_t) = J\nabla H(\hat{z}_t) + e(\hat{z}_t)$, we have

$$H(\hat{z}_T) - H(z_T) = \int_0^T \nabla H(z)^\top \frac{d\hat{z}}{dt} dt = \int_0^T \nabla H(\hat{z}_t)^\top \hat{F}(\hat{z}_t) dt$$

$$= \int_0^T [\nabla H(\hat{z}_t)^\top J\nabla H(\hat{z}_t) + \nabla H(\hat{z}_t)^\top e(\hat{z}_t)] dt$$

$$= \int_0^T \nabla H(\hat{z}_t)^\top e(\hat{z}_t) dt.$$

Bounding the maximum value of the integrand along the path, we have that

$$|H(\hat{z}_T) - H(z_T)| < T\delta \sup \|\nabla H\|, \tag{8}$$

which grows only linearly with time.

This linear bound on the energy error is in stark contrast with the state error which could grow exponentially according to the Lyapunov exponents, even if the dynamics error is arbitrarily small. Advancing the ground truth and learned dynamics forward to some small time $t = \epsilon$, $\hat{z}_\epsilon = z_0 + \epsilon\hat{F}(z_0)$ yields error $\|\hat{z}_\epsilon - z_\epsilon\| = \|\epsilon\hat{F}(z_0) - \epsilon F(z_0)\| = \epsilon\|e(z)\| < \epsilon\delta$. And yet, this error even if propagated by the ground truth dynamics will grow exponentially $\|\hat{z}_T - z_T\| \approx e^{\lambda T}\|\hat{z}_\epsilon - z_\epsilon\| = e^{\lambda T}\epsilon\|e(z)\|$

## B.2  HNN ENERGY CONSERVATION

A simple but erroneous argument for why HNNs approximately conserve the true energy goes as follows:

We would like to know if HNNs achieve better energy conservation given the same levels of error in the predicted dynamics. For HNNs, $\hat{F} = J\nabla\hat{H}$, and we can see that the dynamics error $e(z)$ can also be written as $e(z) = J\nabla(\hat{H} - H)$.

If we could convert a bound on the derivatives $\|e\| = \|\nabla(\hat{H} - H)\| < \delta$ (since $J$ is an orthogonal matrix) into a bound on the learned Hamiltonian itself $E(z) := \hat{H}(z) - H(z) - c$ and $|E| < \Delta$

holding globally for some constant $c$, then we would have a constraint on the energy error that doesn't grow with time. Expanding the difference in initial and final energy, the constant $c$ cancels out and we have

$$H(\hat{z}_T) - H(\hat{z}_0) = \hat{H}(\hat{z}_T) - \hat{H}(\hat{z}_0) - E(\hat{z}_T) + E(\hat{z}_0)$$
$$= -E(\hat{z}_T) + E(\hat{z}_0),$$

using the fact that the learned energy function $\hat{H}$ is conserved over the model rollout $\hat{z}_t$. If there was a constraint $|E| < \Delta$ then

$$|H(\hat{z}_T) - H(\hat{z}_0)| < 2\Delta.$$

Unfortunately, even if the gradients are close and $\delta$ is small, that does not imply that $\Delta$ is small. Small differences in gradient can add up to very large differences in the values of the two functions. While the dynamics may well approximate the data, and achieve low rollout error, there is no reason to believe that at a given point in phase space the learned Hamiltonian should have a value that is close to the true Hamiltonian.

### B.3 SYMPLECTICITY

Symplecticity is the requirement that the dynamics satisfy $(JDF)^\top = JDF$. Defining $G = JF$, the requirement is simply that the antisymmetric part of the jacobian is 0, $DG^\top - DG = 0$.

Unpacking Poincare's lemma requires some familiarity with differential geometry concepts such as differential forms and exterior derivatives, and so we will assume them but for this section only. Poincare's lemma states that on a contractible domain (such as $\mathbb{R}^n$) if a differential $k$-form $\omega$ is closed $d\omega = 0$ (the exterior derivative of $\omega$ is 0) then it is also exact $\omega = d\nu$ (it is the exterior derivative of another differential $(k-1)$-form $\nu$). While $F$ is a vector field, $G = JF$ is a differential 1-form (dual to a vector field). If $DG$ is symmetric, then it is also closed: $dG = \sum_i \partial_i G_j dx^i \wedge dx^j = 2\sum_i (\partial_i G_j - \partial_j G_i) dx^i \wedge dx^j = 0$ since $(\partial_i G_j - \partial_j G_i) = 0$ is just another way of expressing $DG^\top - DG = 0$. Therefore by Poincare's lemma, $G = d\phi$ for some 0-form (scalar function) $\phi$. Therefore $F = J^{-1} d\phi = Jd(-\phi)$ since $J^{-1} = -J$. As the exterior derivative of scalar function is just the gradient, we can define $H = -\phi$ and see that there exists a scalar function $H$ such that $F = J\nabla H$.

## C  MUJOCO EXPERIMENT DETAILS

**Data collection:** for each control task we trained a standard soft actor-critic RL agent to convergence (Haarnoja et al., 2018). Note that we had to use modified versions of the Gym environments since the standard environments preprocess observations in ad-hoc ways. For example, Hopper clips the velocity observations to $[-10, 10]^d$ and truncates part of the position.[2] Our versions of the environments simply return $[q, v]$ as the observation. Then we randomly split the episodes in the replay buffer into train and test. The training data was 40K 3-step trajectories (i.e. two transitions) randomly sampled from the training episodes. The test data was 200 200-step trajectories randomly sampled from the test episodes. This data-collection strategy is important to the experiment because random controls typically do not cause the agent to cover the entire state-action space. Similarly many control policies are highly cyclical, so it is important to separate train and test splits at the episode level.

**Training:** we trained each model for 256 epochs using Adam with a batch size of 200 and weight decay ($\lambda = $ 1e-4). We used a cosine annealing learning rate schedule, with $\eta_{\max} = $ 2e-4, $\eta_{\min} = $ 1e-6.

**Model Architecture** Each network was parameterized as a 2-layer MLP with 128 hidden units. Each model used the Euler integration rule with 8 integration steps per transition step. The step size was determined by the integration step size of the underlying environment, $h = \Delta t/8$.

**NODE + SO:** given the state $z$ and controls $u$, a standard NODE takes $dz/dt = f(z, u, \theta)$. However, if $z = [q, v]$ (that is, if both position and velocity are observed), then we already have a good estimate

---

[2]https://github.com/openai/gym/blob/master/gym/envs/mujoco/hopper.py#L31

of $dq/dt$ in the observation itself, namely $v$. Hence we propose only using the network to model acceleration $dv/dt = f(z, u, \theta)$, and to model $dq/dt$ implicitly. It is important to note that we cannot take $dq/dt = v$ because $v$ is observed before the control $u$ is applied. Instead we take $dq/dt = v/2 + (v + h \times dv/dt)/2$, averaging the velocity at time $t$ (before the control is applied) and the predicted velocity at time $t + 1$ (after the control is applied), given an Euler integration step of size $h$ on $v$.

This integration rule can be viewed as an approximate RK2 step on $q_t$, where $v_{t+1}$ is approximated via a learned Euler step on $v_t$. This approach has two benefits. In the first place it constrains the predicted velocity and acceleration to be consistent across time. Second the model is able to take an approximate RK2 step on $q$ at the cost of a single forward pass (instead of 2). The latter is important because integration error can accumulate over long rollouts, even if the model fits the dynamics very well.

## D  ADDITIONAL EXPERIMENTAL RESULTS

### D.1  COMPARISON OF LOSS FUNCTIONS

In the experimental results presented in the body of the paper were obtained training on l2 loss between integrated and ground truth trajectories. As noted in feedback the an early version of the paper, this practice goes contrary to prior work using l1 loss for stability (Finzi et al., 2020). In Figure 8 we show the result of changing the loss.

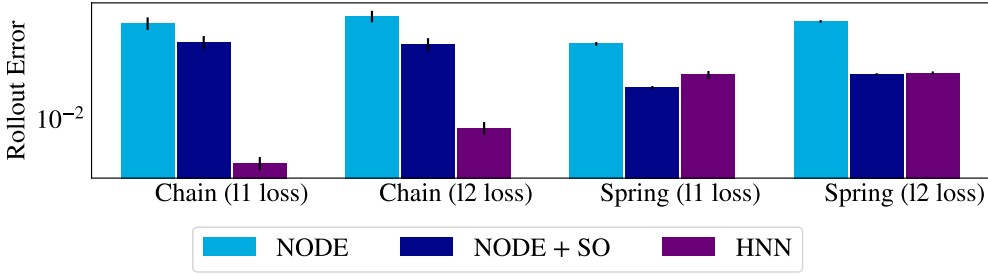

Figure 8:  Switching from l2 to l1 loss can improve rollout error slightly, but doesn't impact the ordering of the models. The other elements of the experimental setup are identical to above. Error bars show one standard deviation.

### D.2  ADDITIONAL SYSTEMS

To extend the comparison of NODE and HNN models, we trained models on three additional systems presented by Finzi et al. (2020). Figure 9 shows the rollout error of NODE, NODE + SO, and HNN models. One the gyroscope system, we observe a similar result to the one above. In the magnet pendulum and rotor systems, the results are slightly more counterintuitive, with the NODE model outperforming the more sophisticated alternatives. We suspect the small difference in performance in the models is due to the challenge of stably training HNNs in complex systems (with magnet pendulum having complex dynamics and rotor having a complex coordinate system).

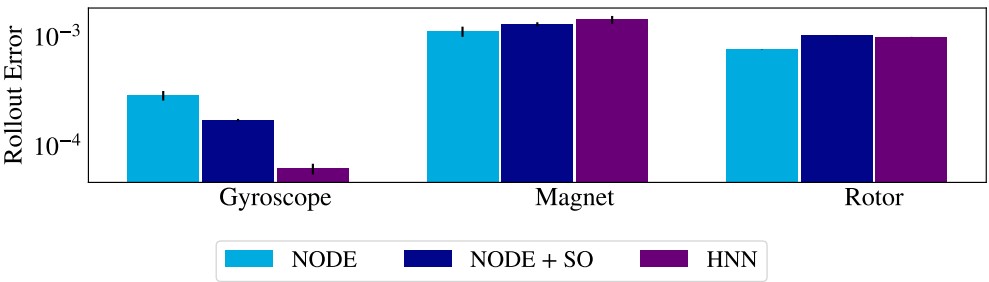

Figure 9: On the additional systems from Finzi et al. (2020), we can observe the effect of second order structure, compared with NODE and HNN baselines. As before, second order structure seems to account for much of the difference between NODE and HNN models. Error bars show one standard deviation.

