# OpenReview forum: "Deconstructing the Inductive Biases of Hamiltonian Neural Networks"
_ICLR.cc/2022/Conference — ICLR 2022 Spotlight_

### Official Review · Reviewer_nLbj · 2021-11-02

**Correctness:** 3
**Technical Novelty And Significance:** 4
**Empirical Novelty And Significance:** 4
**Recommendation:** 8
**Confidence:** 4

**Main Review:**

## Strengths.
The submission is well-motivated and the investigation on HNNs inductive biases is thoroughly performed.
Up to my knowledge, the contributions of this submission are indeed novel.
This submission is definitely relevant for the community as it gives practical insights on commonly encountered inductive biases in physically informed machine learning models.
The writing is also good, making it easy for the reader to follow the narrative.

## Weaknesses.
One weakness perhaps is the lack of further investigation into the failure of HNNs on the MuJoCo physics simulator in Section 5 cf my comments at the end of this review.

## Clarity.
I found the submission well written, with the motivation, ideas and empirical findings well conveyed to the reader. As a consequence, reading this submission was very pleasant.

I would perhaps suggest to add a causal graph---at the end of Section 1 or the begging of Section 4---showing the relationship between the inductive biases (e.g. symplectic dynamics <=> Hamiltonian system, etc). Then it would be easy to understand which assumptions are the strongest etc.

## Relation to prior work.
Related work is discussed properly up to my knowledge.

## Additional feedback.
- Would suggest numbering more equations (at least for the submission).
- "Although energy conservation may be helpful for generalization, the evidence does not indicate that HNNs are better at controlling energy violation than NODEs" -> Perhaps I am missing something, but Figure 2 seems to show that although both HNN and NODE (energy) error grows somewhat linearly wrt rollout time, the HNN's energy error is about ~x10 smaller than NODE's one. Yet, it is indeed interesting to highlight that empirically the energy is not preserved.
- Section 4.2:
    - It is a bit unclear what the vector field F is at first, although it becomes clear a bit later with Equation 1.
    - I would suggest to add a Figure showing phase portraits for a symplectic and a non symplectic system, to give additional intuition to readers that may not be familiar with this concept.
    - Figure 3: Why not having similar plots that in Figure 2? Is the empirical symplecticity growing linearly with rollout time?
- Section 4.3:
    - Would suggest recalling the Hamiltonian dynamics $\frac{d}{dt}[q p]$.
    - $\frac{dp}{dt}=A_\theta(q, p)$ potentially slightly confusing as the second-order structure above is on $v$ with $\frac{dv}{dt}=A(q, v)$. Would advise writing both in terms of $v$ or $p$, or at least reminding the relationship between $p$ and $v$.
- Section 4.4.1:
    - Is the drag term added to all models?
    - What if the drag term is not known?
    - "we show that simply using the bias of second-order dynamics is sufﬁcient to achieve nearly the same performance with much less complexity" -> How come with much less complexity? In what sense?
- Section 5:
    - The subpar performance of HNNs is definitely interesting, but with no further investigation on the underling reasons we are left a bit unsatisfied. Does the MuJoCo physics simulator indeed conserve energy? Is it an Hamiltonian system? Would be interested to have a similar plot than Figure 2, perhaps the true energy is actually not well learnt.


**Summary Of The Paper:**

## Summary and contributions.
Motivated by the success of physics-inspired neural networks, authors investigate the different inductive biases (implicitly or explicitly) encoded in Hamiltonian neural networks (HNNs). Authors identify four biases: the ODE, second order, energy conservation and symplectic biases. They claim and empirically show that, as opposed to what was previously argued, HNNs' performance is mostly due to the implicit second-order bias. Indeed, the authors show that explicitly splitting the Hamiltonian into a momentum and potential term, leads to the system's position $q$ being parametrised as the solution of a second-order differential equation (which involves the mass matrix and the potential function).

**Summary Of The Review:**

Overall this submission is well motivated, thoroughly executed and well written, hence my recommendation for acceptance.

---

> ### Author Response · Authors · 2021-11-15
> **Author Response**
>
> Thank you for your thoughtful comments and feedback.
>
> We will add an additional diagram providing intuition about inductive biases of the model.
>
> In Figure 2, it’s true that the HNN energy error is about 10x smaller than for NODEs; however, the left panel of figure 2 helps explain why this is the case. This panel shows that with HNNs, NODEs, and NODE+SO across the environments, the energy error is best predicted by the state prediction error, rather than the model type. Here HNNs outperform NODEs on predictions (which we later show is a result of the SO bias and the lower complexity from coordinates), and as a result it also has lower energy error. This gap in energy conservation is partially closed by using the NODE+SO model.
>
> Regarding your additional comments:
> * Section 4.2: These are good suggestions, we will clarify that F(z)= dz/dt is the dynamics vector field and perhaps add a phase portrait to the appendix. We would be surprised if the symplectic regularized NODEs did have a symplectic error that consistently grows throughout the trajectory since the regularizer is applied to all points along the training trajectories, however it is an interesting question and we will test this to make sure.
> * Section 4.3: Noted. We will add some clarification about how the (p,q) differential equation is transformed to a (\dot{q},q) differential equation from the relation p=M(q)\dot{q}.
> * Section 4.4.1: Regarding drag, there is a constant drag term added to all of the systems. It is taken to be simply a negative force proportional to the velocity. It is unknown to the models and is learned implicitly by them. Your other question about complexity is a very good one. We refer to complexity in the modeling procedure. In HNNs the dynamics are defined by the gradient of the neural network not simply by its output, and this gives rise to a different set of challenges. However, in a different sense, the complexity of HNNs is lower than NODE + SO because often the functional form of the Hamiltonian is much simpler than the functional form of the vector field. Therefore HNNs can often learn simpler functions. We were referring to the complexity entailed by HNNs in the learning process, but it is important to understand both possible forms of complexity.
> * Section 5: We need to clarify two important things. Firstly, the HNNs/SymODENs described in section 5 will not conserve energy in the same way as vanilla HNNs. As described at the beginning of section 4.4.1, we have added a forcing term to the Hamiltonian. This forcing can add or remove energy from the system. Secondly, MuJoCo is not a conservative system when viewed from the perspective of the agent’s state (which is all that we attempt to model). In the simulator the agent has contacts with the floor that can dissipate energy and which pose a major challenge to simple physics-informed approaches that might rely on smooth dynamics to function well. We could in theory measure the energy in the agent’s state at any given time, but the addition of the forcing term will make direct comparison with the learned Hamiltonian more challenging.

---

> > ### Comment · Reviewer_nLbj · 2021-11-27
> > **(Response)^2**
> >
> > I thank the authors for taking the time to address my concerns and comments.
> > I hope they will indeed update their submission so as to include the clarifications they have made in their responses, and also to add some of the proposed plots / diagrams to improve clarity.
> > I stand by my original score.

---

### Official Review · Reviewer_7KKB · 2021-11-02

**Correctness:** 4
**Technical Novelty And Significance:** 3
**Empirical Novelty And Significance:** 4
**Recommendation:** 8
**Confidence:** 4

**Main Review:**

##########################################################################

 Pros:

-The presentation is clear and easy to follow, even for readers without an extensive mathematical background or a prior familiarity with Hamiltonian dynamics.

-The inductive biases chosen to study are well-motivated. The experiments and analysis used to examine each inductive bias are well-thought-out and the conclusions are supported by the provided evidence.

-The distilled architecture for modeling dynamical systems without the “bells and whistles” of HNNs, NODE + SO, is a solid contribution. This may be surprising to some and seems promising for being used as a baseline in future work. I also found the analysis on the relative energy violation of NODEs vs. HNNs in Section 4.1 compelling and illuminating.

##########################################################################

Cons:

-A takeaway of the experiment in Section 4.2 (Symplectic vector fields) is missing. It would be helpful to add what (if anything) can be concluded by the finding that the symplectic regularizer alone is insufficient to help improve test rollout error.

-For the Mujoco experiment in Section 5, only quantitative results are provided. Given that the page limit is 9 pages but only 8 were used, it seems reasonable to also provide qualitative visualizations that compare rollouts of NODE, NODE + SO, and SymODEN. Providing GIFs/videos would also be helpful for the reader.

-A discussion in Section 5 on why the NODE and NODE + SO Swimmer test time performance differs by a large margin is missing (also, qualitative visualizations could be helpful to explain this).

#########################################################################

Question for authors:

I didn’t quite understand the implications of the Mujoco experiment for the MBRL research community. Is the recommendation that NODE + SO could/should be used as a drop-in replacement for the “simpler dynamics models that predict the next change in state”?  Would it not make sense then to add a baseline to this experiment to compare directly against a simple dynamics model used in a popular Mujoco MBRL framework?

#########################################################################

Some suggestions for improvement:

-In Section 4.1, the line after the equation references “where the *last line* follows” → should be changed to “where the last equality follows”.

-I would suggest defining “symplectic” earlier in the paper, perhaps in the introduction. Currently, it is introduced at the beginning of Section 4.2, although it is discussed at the end of the introduction.

-The last sentence of the introduction, “Extracting the second-order bias, we show how to improve the performance of dynamics models” is unclear to me. Perhaps it could be re-written to more clearly highlight the specific contribution, which to my understanding is: a Neural ODE that models dynamics with second-order information (NODE + SO)?


**Summary Of The Paper:**

This is a well-written analysis of Hamiltonian Neural Networks (HNNs), a class of physics-inspired deep neural networks. The work is motivated by a desire to apply HNNs to non-toy datasets as well as to gain an understanding of the key inductive biases that explain the majority of their performance. Through controlled experiments on synthetic trajectory data, they explore energy conservation, a symplectic bias, complexity of state representation, and second-order structure bias. The key finding is that the second-order structure in HNNs is the main explainer of its performance. A simpler model that combines a Neural ODE with second-order structure is introduced, which can be seen as a distilled HNN. The simpler model achieves stronger performance on Mujoco rollout prediction compared to the HNN.

**Summary Of The Review:**

I think this paper deserves acceptance. In my opinion, it clearly adds new knowledge to the body of work on HNNs, is written exceptionally well, and introduces a useful framework (NODE + SO) for modeling dynamics in non-trivial environments that has the potential to benefit the model-based reinforcement learning (MBRL) community. A few minor requests are provided to strengthen the paper.

==========================================================================

Update after rebuttal: I will be maintaining my initial score and hope that the authors will update their paper for the final camera-ready version to include the promised improvements.

---

> ### Author Response · Authors · 2021-11-15
> **Author Response**
>
> We really appreciate your supportive and thoughtful comments. It means a lot to us.
>
> Regarding your questions:
> * Figure 3 (right) is helpful in understanding the takeaways of section 4.2. This figure shows that even when the symplectic error is very low (and the symplecticity condition is enforced), there is no consistent and discernible effect on the test performance of the model.  We will clarify in the text.
> * Regarding popular baselines for MBRL on MuJoCo, it is very common to use MLPs (or ensembles of MLPs) that model the change in state directly at each time step (sometimes called a DeltaNN). These models can also be viewed as NeuralODEs integrated with Euler’s method and fixed step size 1. We found that using a NODE with step size determined by the sampling rate of the environment was more effective and that using more advanced solvers (e.g. rk4) also further improved performance beyond the simple NODEs common in MBRL (though at a non-trivial increase to computational cost).
> * We want to highlight two specific implications for MBRL research arising from our findings. First, for relatively simple environments like Swimmer, NODE + SO would be a strong drop-in replacement if the goal is to maximize sample-efficiency (as opposed to minimizing compute requirements given some sample-efficiency threshold). Second, since the gains associated with adding SO structure are more muted in more difficult environments, we posit that the MBRL community would benefit more from more work focusing on modeling contacts in relatively stiff systems (e.g. Neural Event Modeling). Furthermore, work such as Zhong et al (2020) could potentially be simplified to keep only SO structure and differentiable contact models, while dropping unhelpful biases and limitations associated with HNNs.
> * On a related note, we conjecture that the large difference in the effect size of adding SO structure between Swimmer and the other environments is likely due to the difference between the nature of the contacts present in HalfCheetah and Hopper environments versus those present in Swimmer. In the former, the contacts are impacts that cause rapid changes in velocity, while in the latter contact primarily results in friction.

---

> > ### Comment · Reviewer_7KKB · 2021-11-28
> > **Response to authors**
> >
> > Thank you for the responses to my feedback and questions! I will be maintaining my original score.
> >
> > My hope is that the updated paper will contain the clarifying points made in the author's response, particularly because I believe it will help make the paper more accessible to a wider audience.
> >
> > Specifically:
> > * The promised clarifying point about Figure 3 (right)
> > * The explanation clarifying the connection between common approaches in MBRL and the NODE + SO, e.g., *"We found that using a NODE with step size determined by the sampling rate of the environment was more effective and that using more advanced solvers (e.g. rk4) also further improved performance beyond the simple NODEs common in MBRL (though at a non-trivial increase to computational cost)."*
> > * The two specific implications for MBRL research

---

### Official Review · Reviewer_SW9u · 2021-11-03

**Correctness:** 3
**Technical Novelty And Significance:** 3
**Empirical Novelty And Significance:** 3
**Recommendation:** 8
**Confidence:** 3

**Main Review:**

Strenghts:

This paper tackles an interesting and useful question of introspecting and analyzing different inductive biases. I think this work presents an important analysis which could have useful downstream applications. The experiments and analysis are well written and thorough, and the break down of how each type of inductive bias might help or hurt is interesting and easy to follow. The experiments related to the MuJoCo benchmarks help ground the need for this type of analysis, and open the door to some interesting future work.


Weaknesses:

I think the weakenesses in this paper are quite minor. It would be good to see more empirical analysis on more complex tasks such as the MuJoCo benchmark tasks instead of the kChainPendulum and kSpringPendulum tasks, especially since we see that the biases of HNNs have difficulty with modeling the dynamics for the MuJoCo tasks. A discussion or experimentation on where each of the inductive biases are actually useful would also be interesting.

**Summary Of The Paper:**

This paper presents an in-depth study of the inductive biases in physics-inspired neural networks, especially Hamiltonian Neural Networks (HNNs). The authors break down the biases in HNNs into multiple categories, such as energy conservation of the network, second-order structure of the output, symplectic vector fields produced by the networks, and the role of the complexity of the coordinate system. The authors show through experiments on a set of pendulum-based tasks, that HNNs are not in fact better at conserving energy than competing approaches that dont explictly model this such as Neural ODEs. They also show that Regularizing NODEs with a symplectic field bias does not help in generalization. Through experimentation, they show that a second order output structure similar to that of HNNs helps NODEs quite a bit and outperforms SymODEN. Finally, the authors use these findings to obtain good results in modeling trajectory dynamics with NODEs and a second order output to model dynamics in MuJoCo tasks.

**Summary Of The Review:**

The paper presents a thorough and well written analysis of inductive biases in HNNs.

---

> ### Author Response · Authors · 2021-11-15
> **Author Response**
>
> Thank you for your thoughtful comments and feedback.
>
> While we evaluate on the kChainPendulum (2d with hard constraints) and the kSpringPendulum (3d with no constraints) and the Mujoco environments (3d with constraints and contacts) which have a substantial diversity in the qualities of the dynamics, we appreciate your point that additional diversity would add strength to the paper. In order to address this, we plan to add evaluations on the Gyroscope, Magnet Pendulum, and Rigid Rotor 3d systems from Finzi et al. 2020 to our experiments.

---

> > ### Comment · Reviewer_SW9u · 2021-11-29
> > **Thanks for the response!**
> >
> > Thank you for your response!

---

### Official Review · Reviewer_wu5x · 2021-11-05

**Correctness:** 3
**Technical Novelty And Significance:** 3
**Empirical Novelty And Significance:** 3
**Recommendation:** 6
**Confidence:** 4

**Main Review:**

The main contribution and novelty of this work lie in the insight it provides about the role of various, often intertwined, inductive-biases present in the networks that encode Hamiltonian and Lagrangian dynamics into their network structure. Furthermore, through carefully motivated and executed experiments, it shows that incorporation of second-order structure into the learning framework is the most important factor in improving the performance of Hamiltonian/Lagrangian dynamics based neural networks.

Following are some more specific concerns/comments:

-- While showing the relationship between the accuracy of energy conservation and rollout error in Figure 2, the authors have not provided a clear definition of *energy violation*. Please clearly define this performance metric in terms of $H$ and $\hat{H}$.

-- On a related note, the derivation and motivation of the inequalities in Section A.2 are not very clear. The constant present in the inequality "$|\hat{H} - H - const|<\Delta$" represents the amount of mismatch between the learned and the predicted Hamiltonian that can be attributed to the change of reference in potential energy. However, this quantity has not been connected with the rest of the derivation. Also, the last sentence of Section A.1 does not immediately follow the previous discussions. Can one use Gronwall–Bellman inequality) to make this claim about exponentially growing state error? It would be much appreciated if these theoretical discussions were made more precise.

-- This paper shows that HNN/SymODEN performs poorly in control-related tasks from MuJoCo (e.g., HalfCheetah or Hopper). This is not unexpected since HNN/SymODEN tries to conserve energy. At the same time, Neural ODE (with or without Second-Order bias) does not have any such inclination, and energy may not be a conserved quantity in the ground-truth data (presence of contacts and collisions can often change the total energy of the system). Therefore, this does not appear to be a fair comparison. On the other hand, recent work by Zhong et al. (*Extending Lagrangian and Hamiltonian Neural Networks with Differentiable Contact Models*, arXiv:2102.06794) has extended Hamiltonian dynamics based networks to systems with contacts and collisions. I would encourage the authors to use this approach as one of the baselines while running the experiments for Section 5.

-- Finzi et al. (*Simplifying Hamiltonian and Lagrangian Neural Networks via Explicit Constraints*, NeurIPS 2020)have shown that the $L_1$ loss functions exhibit more robustness in these problems. Can the authors confirm if the conclusions of this paper hold when the loss function is changed from $L_2$ to $L_1$?

-- Also, it is not clear how the relative performance of the different models changes with an increase/decrease in the neural network parameters. I would strongly encourage the authors to explore these directions.

**Summary Of The Paper:**

Incorporating physics-informed inductive bias, especially Hamiltonian and Lagrangian dynamics, into deep neural networks has been the focus of a fast-growing body of work over the last few years. This paper has carried out a formal analysis of these methods to identify which specific aspect(s) of these energy-conserving networks contribute(s) the most to their superior performance in prediction and generalization. Through theoretical and empirical analyses, this paper concludes that incorporating the second-order structure (i.e., using a neural network to model the acceleration) and reducing functional complexity through a change of coordinates, not energy conservation or the symplectic structure, play the most critical role in improving the performance. The experiments are thorough and emphasize the message precisely.

**Summary Of The Review:**

This paper has looked into the existing solution approaches to a relevant and interesting problem and focused on understanding how those solution approaches work. This analysis provided some insight which, in turn, were exploited to introduce a new architecture that can match/outperform the existing methods with much simpler network architecture. The idea is interesting, the experiments are mostly well carried out, and the paper is reasonably well-written. However, as explained in the *main review*, some aspects should be further improved to increase the concreteness and overall clarity of the paper.

---

> ### Author Response · Authors · 2021-11-15
> **Author Response**
>
> Thank you for your thoughtful and supportive comments.
> For energy conservation, we are plotting the quantity $\|H(\hat{z}_t)-H(z_t)\|/\|H(\hat{z}_t)\|\|H(z_t)\|$, measuring the relative error of the true energy function evaluated on the rollout trajectory $\hat{z}_t$ as compared to the ground truth trajectory $z_t$. Thanks for the question, we clarify in the text.
> * Thanks for your question about l1 vs l2 loss. Inspired by your comments, we trained with the l1 loss on chain and spring pendulum systems with 3 bodies, and found the overall ranking of methods in each case to be unchanged. Moving from l2 to l1 loss led to a mild degradation of performance of HNNs and NODE + SO (3% and 10% in the geometric mean of relative error over 5 seeds) on the spring pendulum, and a mild improvement for NODE (12%). On the chain pendulum, the l1 loss led to improvements for the HNN and NN (22% and 10%), and essentially no change for the NODE + SO. We will include these results in detail in the Appendix.
> * Inequalities in appendix A.2: Thanks for the questions. To clarify, we meant to say that if $\|\hat{H}(z)−H(z)−const\|<\Delta$, then it would follow that $|H(\hat{z}_T ) − H(\hat{z}_0)| < 2\Delta$ (the const cancels as it appears on both sides of the expression). In the last sentence of A1 we meant that if the underlying dynamics were the same $\hat{F}(z)=F(z)$, small discrepancies in the state can grow exponentially according to the maximal Lyapunov exponent $\lambda$, $\|\hat{z}_t-z_t\| \approx e^{\lambda t}\|\hat{z}_\epsilon-z_\epsilon\|$. Since the learned dynamics differ slightly from the ground truth, the discrepancy between the trajectories could grow perhaps even at a larger rate. We will clarify these points in the appendix.
> * Regarding more sophisticated baselines, as far as we know, with the additional forcing term SymODEN does not conserve energy. But we appreciate your point that this model may have difficulty learning various kinds of dissipation and contacts, and we will add the method from Extending Lagrangian and Hamiltonian Neural Networks with Differentiable Contact Models to the baselines for the camera ready version of the paper.
> * HNNs typically require fewer parameters than NODEs and NODEs + SO, and can also more easily overfit with a smaller number of parameters. In many cases, the functional form of the Hamiltonian is much simpler than the functional form of the dynamics, so it can be more easily found by a small MLP. Our focus here, however, is primarily in understanding the predictive performance of HNNs, which we show is mostly explained by simple second order inductive biases.

---

> > ### Comment · Reviewer_wu5x · 2021-11-28
> > **Reply to Author Response**
> >
> > I would like to thank the authors for responding to my concerns/comments. I expect that they will incorporate the changes in the final manuscript.

---

### Author Response · Authors · 2021-11-15
**Summary Note**

Identifying which inductive biases are relevant to predictive performance and which are not is crucial for simplifying our machine learning models and designing better ones. Despite the conventional narrative around Hamiltonian Neural Networks, we show that energy conservation and symplecticity have little to no impact on the predictive performance of these models and that instead second-order structure accounts for a large amount of their improved performance. By stripping away the unnecessary energy conservation and symplecticity properties, we are able to apply our model to dynamical systems that would violate these properties, such as the Mujoco locomotion environments. The reviewers are unanimously supportive of our paper, which they found to be insightful about the inner workings of HNNs, and clearly written. Below we respond to reviewer questions individually.

---

### Decision · Program_Chairs · 2022-01-20

**Decision:**

Accept (Spotlight)

**Comment:**

This paper examined physics-inspired inductive biases in neural networks, in particular Hamiltonian and Lagrangian dynamics. The work separated the benefits arising from incorporating energy conservation, the symplectic bias, the coordinate systems, and second-order dynamics.  Through a set of experiments, the paper showed the most important factor for improved performance in the test domains was the second-order dynamics, and not the more common explanation of energy conservation or the other factors. The increased generality of this approach was demonstrated with better predictions on Mujoco tasks that did not conserve energy.

All reviewers liked the insights provided by the paper.  They agreed that the paper clearly laid out several hypotheses and systematically tested them.  The reviewers found the experiments thoughtful and the results compelling.  The reviewers also pointed out several aspects of the document that could be improved, including additional formalism clarifications (reviewer nLbj), baseline algorithms (reviewer wu5x), and domains (reviewers 7KKB,SW9u).  The reviewers found the author's response satisfactory but were disappointed that a revised paper was not ready to read. The reviewers want the final paper to include the modifications that were promised in the author response.

All four reviewers indicated to accept this paper which contributes novel insights that simplify and generalize physics-inspired neural networks. The paper is therefore accepted.